# KVSharer: Efficient Inference via Layer-Wise Dissimilar KV Cache Sharing

## Abstract

The development of large language models (LLMs) has significantly expanded model sizes, resulting in substantial GPU memory requirements during inference. The key and value storage of the attention map in the KV (key-value) cache accounts for more than 80% of this memory consumption. Nowadays, most existing KV cache compression methods focus on intra-layer compression within a single Transformer layer but few works consider layer-wise compression. In this paper, we propose a plug-and-play method called *KVSharer*, which shares the KV cache between layers to achieve layer-wise compression. Rather than intuitively sharing based on higher similarity, we discover a counterintuitive phenomenon: sharing dissimilar KV caches better preserves the model performance. Experiments show that *KVSharer* can reduce KV cache computation by 30%, thereby lowering memory consumption without significantly impacting model performance and it can also achieve at least 1.3 times generation acceleration. Additionally, we verify that *KVSharer* is compatible with existing intra-layer KV cache compression methods, and combining both can further save memory.

## 1 Introduction

Recently, large language models (LLMs) built on the Transformer (Vaswani et al., 2017) architecture have demonstrated remarkable abilities across a wide range of tasks (Touvron et al., 2023; Cai et al., 2024; Yang et al., 2024a; Brown, 2020; Jiang et al., 2023). However, these impressive capabilities usually come with a significant increase in model size, resulting in substantial GPU memory costs during inference. The memory consumption of LLM during inference primarily comes from model parameters and the KV (key-value) cache. The KV cache is a commonly used technique in the efficient inference of LLM, which stores the keys and values previously computed in the attention mechanism, allowing for reuse in subsequent generation processes to improve inference speed. Although the KV cache greatly helps improve inference speed, it also significantly pressures memory usage. During the LLM inference phase, the KV cache typically accounts for 80% of the total memory usage, making it essential to optimize the KV cache to reduce memory consumption (Yang et al., 2024b; Zhang et al., 2024b).

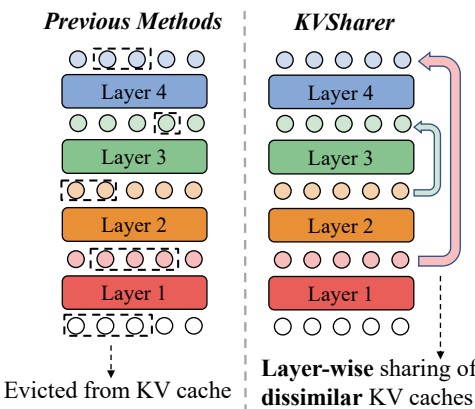

Figure 1: Previous methods primarily focus on discarding Keys and Values within layers. In contrast, we share KV caches across layers based on their dissimilarity, where dissimilarity refers to the dissimilarity of vectors formed by flattening the KV caches.

Recent research has seen a proliferation of methods aimed at compressing KV caches to reduce memory consumption (Zandieh et al., 2024; Xu et al., 2024; Yang et al., 2024b; Zhang et al., 2024b;a; Dong et al., 2024). However, these efforts have predominantly focused on intra-layer KV cache compression within individual Transformer layers of LLM. In contrast, layer-wise KV cache compression strategies, which calculate the KV cache for only a subset of layers to minimize

memory usage, remain largely unexplored. The limited existing work on layer-wise KV cache compression typically requires additional training to maintain satisfactory performance (Wu & Tu, 2024; Liu et al., 2024a).

In this paper, we introduce *KVSharer*, a plug-and-play method for compressing the KV cache of well-trained LLMs. Contrary to the intuitive expectation of sharing similar KV caches, i.e., the vectors formed by flattening KV caches being highly identical, our method is based on an empirically discovered counterintuitive phenomenon: when the KV caches of two layers differ significantly, sharing one layer's KV cache with another during inference does not lead to significant performance degradation. The paradox in this discovery lies in that previous methods for sharing parameters or activation values have always relied on replacing similar values (Dehghani et al., 2018; Reid et al., 2021; Cao et al., 2024). In contrast, we are the first to show that, in the context of KV caches, model performance can be effectively maintained by sharing dissimilar layer-wise KV caches. Leveraging this observation, *KVSharer* employs a search strategy to identify the KV cache sharing strategy across different layers during inference. *KVSharer* significantly reduces GPU memory consumption while maintaining most of the model performance. For example, it retains over 95% of the model performance while using only 70% of the original memory. As a layer-wise KV cache compression technique, *KVSharer* is compatible with existing intra-layer KV cache compression methods, offering a complementary approach to memory optimization in LLMs. *KVsharer* is also a general method and not task-specific, meaning that once a sharing strategy is found on a general calibration dataset, it can be directly applied to any downstream task. Our contributions are summarized as follows:

- We first discover a counterintuitive phenomenon where sharing dissimilar KV caches does not significantly degrade model performance. Based on this, we introduce *KVSharer*, a layer-wise KV cache sharing mechanism for efficient inference without additional training.

- Experiments using PPL (Perplexity) and various downstream benchmarks demonstrate that *KVSharer* can effectively reduce memory consumption without significantly affecting model performance. *KVSharer* also has the effect of improving generation speed.

- *KVSharer* is compatible with the current intra-layer KV cache compression methods, enabling further memory reduction while maintaining good model performance.

## 2 RELATED WORK

### 2.1 KV CACHE COMPRESSION

Most of the existing KV cache compression work is carried out within a single transformer layer, namely the intra-layer compression. For example, StreamingLLM (Xiao et al., 2023) only retains the attention sink in the KV cache, avoiding a significant increase in memory demand when generating long texts. H2O (Zhang et al., 2024b) reduces memory usage by removing the keys and values stored by unimportant tokens from the full KV cache. Compared to H2O, Scissorhands (Liu et al., 2024b) discards as many tokens as possible from the KV cache in each round, rather than just one token. PyramidInfer (Yang et al., 2024b) considers calculating the key-values only for important tokens during generation. FastGen (Ge et al., 2023) also discards the attention values of certain non-special tokens in the KV cache but sets a maximum approximation error for the attention matrix to ensure model performance. SnapKV (Li et al., 2024) builds on the observation that attention heads tend to consistently focus on certain prompt features, especially those toward the end, to compress KV caches by selecting key positions for each head. While these methods have shown effective compression ability, they achieve KV cache sparsification by discarding tokens within a single layer. However, they do not address layer-wise KV cache compression.

Recently, only a few works have focused on layer-wise compression strategies for the KV cache. MiniCache (Liu et al., 2024a) merges the KV caches from different layers to enhance throughput. LCKV (Wu & Tu, 2024) proposes a novel method that computes and caches the KVs for only a small number of layers, thereby significantly reducing memory consumption and improving inference throughput. CLA (Brandon et al., 2024) design an inter-layer attention mechanism to share the KV cache across different layers. YOCO (Sun et al., 2024) designs a decoder-decoder architecture that enforces the reuse of the lower layer's KV cache in the higher layers' KV cache. However, all of them require further training of the model rather than being plug-and-play on well-trained LLMs.

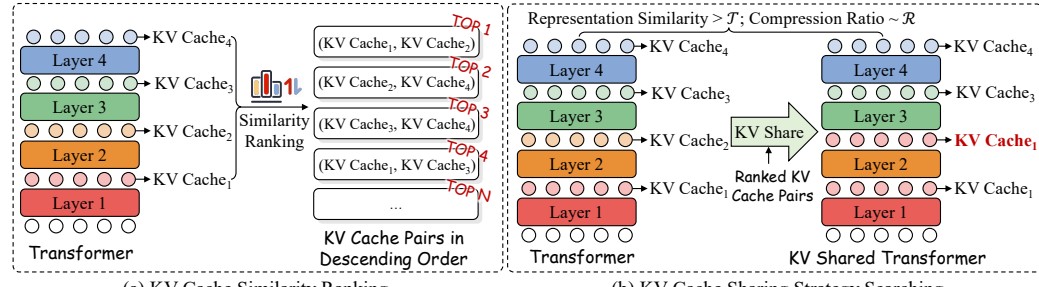

(a) KV Cache Similarity Ranking      (b) KV Cache Sharing Strategy Searching

Figure 2: An illustration of the strategy searching process of the *KVSharer*. For a given LLM, process (a) performs inference on the calibration dataset and computes the euclidean distance between flattened KV cache vectors from any two layers, sorting pairs in descending order. (b) KV cache pairs are sequentially replaced, ensuring the final hidden-state similarity with the original model exceeds threshold $\mathcal{T}$ until the KV cache compression ratio reaches $\mathcal{R}$.

In contrast, we are the first to propose a layer-wise KV cache compression method for well-trained LLMs without further training. Moreover, our method is directly compatible with the current intra-layer KV cache compression techniques.

## 2.2 ATTENTION MAP & PARAMETER SHARING

Since the introduction of Transformer-based pre-trained language models (PLMs) like BERT (Devlin et al., 2018), some research has focused on attention map sharing and parameter sharing. Lazyformer (Ying et al., 2021) reuses attention maps from lower layers in higher layers of the Transformer, thereby enhancing the throughput of PLMs. Xiao et al. (2019) directly share the attention weights across layers, improving inference speed in machine translation tasks. Takase & Kiyono (2021) design three parameter sharing strategies based on rules within the Transformer architecture, improving model efficiency in machine translation tasks. Shim et al. (2023) conduct a comprehensive evaluation of various attention map sharing strategies. Since the advent of the era of LLMs, various works utilizing parameter sharing or attention map sharing have been widely adopted. Multi-Query attention (MQA) (Shazeer, 2019) and Grouped-Query attention (GQA) (Ainslie et al., 2023) have become standard strategies in modern LLMs, improving model efficiency by sharing attention queries and keys within a layer. Cao et al. (2024) investigate the similarity of attention maps and attention parameters in LLMs and propose various attention map sharing strategies to reduce inference memory consumption. However, none of these works have extended to the KV cache. They all rely on replacing layers with higher parameter similarity or activation values, which aligns with intuition, whereas we replace dissimilar KV cache.

## 3 KVSHARER

The main steps of *KVSharer* are divided into two parts. First, for a given LLM, it searches a sharing strategy, a list that specifies which layers' KV caches should be replaced by those of other specific layers. Then, during the subsequent prefill and generation processes on all the tasks, the KV caches of the relevant layers are directly replaced according to this list, enabling efficient inference.

### 3.1 STRATEGY SEARCHING

To heuristically search for a sharing strategy, our approach is to first perform inference on a calibration dataset and calculate the euclidean distance between the KV caches of any two layers. Then, we sort these KV cache pairs in descending order of euclidean distance. Subsequently, we attempt to replace the corresponding KV caches in sequence, while ensuring that the model's output remains as consistent as possible with the original model during the replacement process. The search process can be referenced in Algorithm 1 and Figure 2.

---

**Algorithm 1** Workflow of Strategy Searching

---

**Require:** LLM $\mathcal{M}$, Target Shared KV Cache Layers $\mathcal{C}$, Calibration Dataset $\mathcal{D}$, Threshold for representation similarity $\mathcal{T}$

**Ensure:** Sharing Strategy $\mathcal{Z}$

1:  $\mathcal{S} \leftarrow$ Euclidean_KV_Dis($\mathcal{M}, \mathcal{D}$)     $\triangleright$ Perform inference on the calibration dataset $\mathcal{D}$, compute the euclidean distance between the KV caches of any two layers, and record the corresponding layer pairs and their distance values as $\mathcal{S}$

2:  $\mathcal{R} \leftarrow$ Descend_Rank($\mathcal{S}$)     $\triangleright$ Sort the KV cache layer pairs in descending order based on their euclidean distance

3:  $\mathcal{Z} \leftarrow \emptyset$                             $\triangleright$ Initialize candidate sharing strategy as $\mathcal{Z}$

4:  $\mathcal{P} \leftarrow 0$                             $\triangleright$ Initialize current number of shared layers as $\mathcal{P}$

5: **for** each $r$ in $\mathcal{R}$ **do**

6:     $\mathcal{Z} \leftarrow \mathcal{Z} \cup r$                    $\triangleright$ Add the current pair $r$ to the candidate set

7:     $\mathcal{M}_{tmp} \leftarrow$ Sharing_KV($\mathcal{M}, \mathcal{Z}$)     $\triangleright$ Apply layer-wise KV cache sharing to $\mathcal{M}$ according to the current candidate strategy and get candidate model $\mathcal{M}_{tmp}$

8:     $s \leftarrow$ Avg_Cos_Sim($\mathcal{M}_{tmp}, \mathcal{M}, \mathcal{D}$)   $\triangleright$ Compute the similarity of the final layer hidden-state between the two models on the calibration dataset as $s$

9:     **if** $s <= \mathcal{T}$ **then**

10:       $\mathcal{Z} \leftarrow \mathcal{Z} \setminus r$ $\triangleright$ If the output similarity between the current model and the original model falls below the threshold, the current pair $r$ is discarded

11:     **else**

12:       $\mathcal{P} \leftarrow \mathcal{P} + 1$                 $\triangleright$ Find a replacement and increase the shared layers $\mathcal{P}$ by 1

13:       **if** $\mathcal{P} == \mathcal{C}$ **then**

14:         **return** $\mathcal{Z}$       $\triangleright$ Return the currently found optimal strategy when the number of compressed layers reaches the preset value $\mathcal{C}$

15:       **end if**

16:     **end if**

17: **end for**

18: **return** None

---

### 3.1.1 PREPARATION

For a given LLM $\mathcal{M}$, we set the target number of shared KV cache layers $\mathcal{C}$. We specify a calibration dataset $\mathcal{D}$, which typically consists of several plain sentences. We conduct forward computations on $\mathcal{D}$ using both the model with shared KV cache and the original model to obtain output representations, ensuring that the cosine similarity of these representations exceeds the threshold $\mathcal{T}$.

### 3.1.2 SEARCHING

**KV Cache Similarity Calculation & Initialization (1-4)** First, we perform a forward pass using the original model $\mathcal{M}$ on the calibration dataset $\mathcal{D}$, saving the KV cache for each layer during the forward pass of each sentence. Then, we average the KV cache for each layer across all samples to obtain the average KV cache for each layer. Finally, we flatten the keys and values of the KV cache for each layer into a one-dimensional vector, and then average the keys and values separately to represent the KV cache for that layer. We then calculate the euclidean distance between the KV cache representations of any two layers to obtain $\mathcal{S}$. We then sort $\mathcal{S}$ in descending order to get $\mathcal{R}$, as a larger euclidean distance indicates lower similarity. Consequently, dissimilar layer pairs are prioritized. We then set two variables, $\mathcal{Z}$ and $\mathcal{P}$, to record the candidate KV cache sharing strategy and the current number of shared layers.

**Sharing Strategy Searching (5-18)** Based on the values in $\mathcal{R}$, we sequentially select a pair of layers $r$ to add to $\mathcal{Z}$ for sharing. When sharing, we replace the layer closer to the output with the one closer to the input, as the layers near the input end in LLMs are more sensitive, and modifying them could result in significant performance degradation (Cao et al., 2024; Yang et al., 2024c).

We then apply the candidate strategy $\mathcal{Z}$ by directly replacing the KV cache of one layer with another during the forward pass. Using the model with KV cache sharing and the original model, we perform inference on the calibration dataset to obtain the output representation from the last layer. We

then average these representations across different sentences. If the cosine similarity between the averaged output representations of the two models exceeds the threshold $\mathcal{T}$, we retain the current pair replacement $r$; otherwise, we discard it. This iteration continues until the predefined number of compressed layers $\mathcal{C}$ is reached. At the end of the iteration, we obtain an optimal KV cache sharing strategy $\mathcal{Z}$ through the heuristic search.

## 3.2 INFERENCE WITH KV CACHE SHARING

After obtaining the KV cache sharing strategy $\mathcal{Z}$, we apply it to all subsequent inference tasks, including both prefill and generation processes. As illustrated in Figure 3, during forward computations, when a layer's KV cache needs to be replaced based on $\mathcal{Z}$, we directly copy the KV cache from the previously computed layer. The subsequent computations then follow the original model's process.

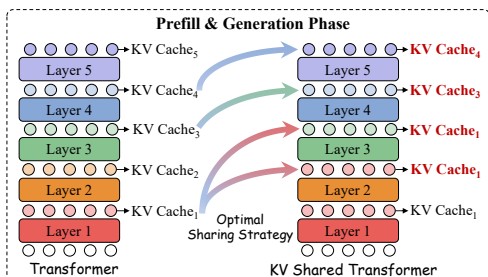

Figure 3: During the inference process of prefill and generation, according to the currently found optimal sharing strategy, *KVSharer* directly copy the result of the KV cache from a previously computed layer to the current layer during the forward computation.

## 4 EXPERIMENTS

### 4.1 MODELS

To evaluate the effectiveness of the proposed *KVSharer*, we perform experiments on widely-used English LLMs, specifically Llama2-7B and 13B (Touvron et al., 2023). We also examine its effectiveness on bilingual LLMs, namely InternLM2-7B and 20B (Cai et al., 2024), which support both Chinese and English. For main experiments, we utilize the chat versions of Llama2-7B, InternLM2-7B, InternLM2-20B and Llama2-13B. We choose these two model series because they offer open-source models in a relatively complete range of different sizes and versions (Base or Chat). Additionally, we include experiments on the advanced Mistral-7B-Instruct-v0.3 (Jiang et al., 2023) to validate the universality of our method.

### 4.2 BENCHMARKS

To comprehensively evaluate the model's widely focused capabilities, we utilize the OpenCompass evaluation framework Contributors (2023). Specifically, we conduct evaluations in five aspects: Reasoning, Language, Knowledge, Examination and Understanding. We select several benchmarks from each category. **Reasoning:** CMNLI Xu et al. (2020), HellaSwag (HeSw) Zellers et al. (2019), PIQA Bisk et al. (2019). **Language:** CHID Zheng et al. (2019), WSC Levesque et al. (2012). **Knowledge:** CommonSenseQA (CSQA) Talmor et al. (2018), BoolQ Clark et al. (2019). **Examination:** MMLU Hendrycks et al. (2021), CMMLU Li et al. (2023). **Understanding:** Race-High/Middle (H/M) Lai et al. (2017), XSum Narayan et al. (2018), C3 Sun et al. (2020). We perform evaluations using the official scripts from OpenCompass, employing zero-shot or few-shot approaches without any additional training. Two evaluation modes are employed: perplexity (PPL) and generation (GEN) [1]. The GEN mode is used for CHID and XSum, while both PPL ($WSC_P$) and GEN ($WSC_G$) modes are applied to the WSC dataset. The remaining benchmarks are assessed using the PPL mode. OpenCompass then converts the evaluation results for each benchmark into a score, with higher scores indicating better performance.

### 4.3 SETTINGS

We configure the compression rates for each model at 12.5%, 25%, and 37.5% by setting the target shared KV cache layers $\mathcal{C}$, as subsequent results show that the models can maintain relatively good performance within this range. For all the models, we randomly select 30 sentences from English Wikipedia as the calibration dataset where each sentence has 64 tokens. We set $\mathcal{T}$ to 0.5 for all

---

[1] https://opencompass.readthedocs.io/en/latest/get_started/faq.html

Table 1: The main results of our experiments. "Layer" represents the number of layers where the KV cache is actually computed. We present the average values of the model across different aspects of tasks and the average scores of all tasks as percentages relative to the full KV cache.

| LLM | Layer | Average | Percent | Reasoning | Language | Knowledge | Examination | Understanding |
|---|---|---|---|---|---|---|---|---|
| Llama2-7B | 32 | 46.55 | 100% | 60.83 | 40.67 | 68.67 | 38.89 | 33.03 |
| | 28 | 52.89 | 113.6% | 60.73 | 54.41 | 71.86 | 36.18 | 44.73 |
| | 24 | 45.58 | 97.9% | 57.74 | 51.00 | 60.52 | 33.12 | 31.15 |
| | 20 | 38.55 | 82.8% | 53.68 | 38.52 | 53.90 | 26.94 | 25.37 |
| Llama2-13B | 40 | 58.13 | 100% | 62.90 | 60.56 | 75.67 | 46.67 | 49.67 |
| | 35 | 56.32 | 96.9% | 61.31 | 57.33 | 74.56 | 46.16 | 47.77 |
| | 30 | 51.97 | 89.4% | 61.35 | 47.38 | 73.66 | 46.03 | 40.51 |
| | 25 | 40.50 | 69.7% | 57.46 | 43.75 | 52.39 | 35.39 | 21.97 |
| InternLM2-7B | 32 | 68.63 | 100% | 62.00 | 71.30 | 76.37 | 64.46 | 69.82 |
| | 28 | 66.57 | 97.0% | 60.81 | 66.99 | 75.32 | 60.26 | 69.36 |
| | 24 | 66.59 | 97.0% | 62.19 | 65.37 | 74.66 | 62.70 | 68.71 |
| | 20 | 65.01 | 94.7% | 61.10 | 63.74 | 74.28 | 62.54 | 65.51 |
| InternLM2-20B | 48 | 70.82 | 100% | 70.66 | 67.34 | 77.88 | 66.26 | 72.33 |
| | 42 | 69.80 | 98.6% | 69.02 | 66.84 | 77.39 | 65.94 | 70.75 |
| | 36 | 68.99 | 97.4% | 66.82 | 65.55 | 77.41 | 65.45 | 70.76 |
| | 30 | 66.96 | 94.5% | 66.58 | 59.62 | 77.41 | 65.07 | 68.48 |
| Mistral-7B | 32 | 64.13 | 100% | 64.78 | 62.76 | 79.03 | 53.49 | 62.53 |
| | 28 | 61.56 | 96.0% | 64.40 | 57.88 | 77.35 | 50.02 | 60.05 |
| | 24 | 56.67 | 88.4% | 63.28 | 50.15 | 76.18 | 45.23 | 52.55 |
| | 20 | 50.53 | 78.8% | 61.40 | 49.72 | 71.50 | 35.50 | 40.02 |

the models [2]. All experiments related to the PPL evaluation are conducted on a Wikipedia dataset consisting of 200 sentences, where the token length of each sentence is set to 2048. We perform experiments on a server equipped with 4 Nvidia A100 80GB GPUs.

## 4.4 MAIN RESULT

We conduct experiments on each dataset, calculate the average score for each aspect, the average score across all tasks, and the percentage of the average score for all tasks using *KVShare* compression relative to the average score with the full KV cache in Table 1. Detailed results can be found in Table 6 of Appendix A.1.

Llama2-7B and InternLM2-7B each have 32 layers, while Llama2-13B and InternLM2-20B have 40 and 48 layers, respectively. To evaluate performance, we apply different numbers of compressed layers to the four models at compression rates of 12.5%, 25%, and 37.5%. Additionally, we include models with full KV cache for comparison. Based on the main results, *KVSharer* exhibits minimal performance degradation compared to

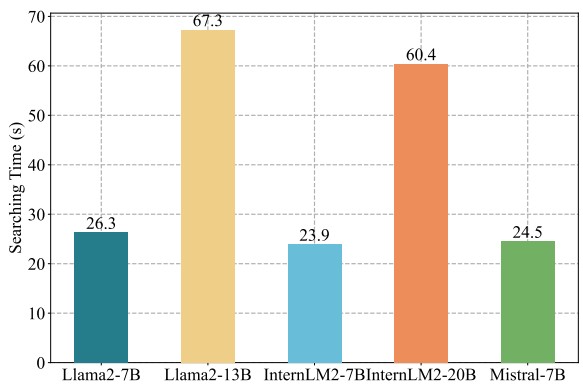

Figure 4: The searching time cost by *KVSharer* for different models. The search time is typically around 60 seconds or less.

the full KV cache in the vast majority of tasks. Notably, when the compression rate is 25% or less, the performance remains close to 90%, and in some cases, even exceeds 95%. Furthermore,

---

[2]When strategy searching, the similarity of the last layer's hidden state between the compressed model and the original model is usually greater than 0.8. We set a threshold of 0.5 to avoid rare cases of model output collapse. Since this situation is infrequent, we do not perform an ablation study on $\mathcal{T}$.

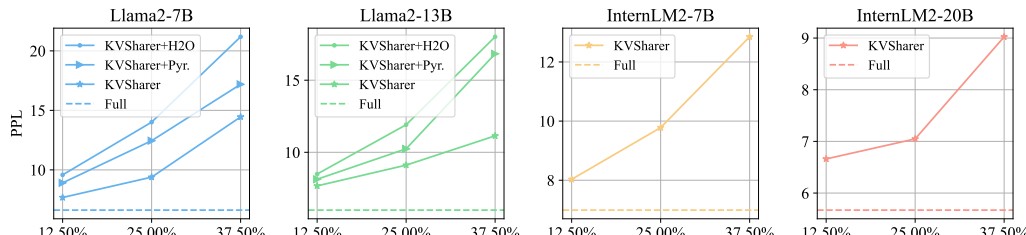

Figure 5: The model's perplexity on the Wikipedia dataset at different compression rates. "+H2O" and "+Pyr." refer to the additional use of the H2O and PyramidInfer for intra-layer compression.

the model does not suffer significant performance drops in any specific aspect, as no individual score approaches zero. These results demonstrate that *KVSharer* effectively preserves the model's overall and task-specific performance. To present the results in Table 1 more intuitively, we show the average performance of each model across all tasks at different compression rates, as illustrated in Appendix A.1 Figure 7. It can also be observed that *KVSharer* can maintain the model's performance well with a compression rate of 25% or less, and even improves the average performance of the model at a 12.5% compression rate on Llama2-7B.

We also validate the larger Llama2-70B model using several benchmarks and PPL, discovering that *KVSharer* is also effective for it, maintaining most of its performance, as detailed in Appendix A.2.

### 4.5 STRATEGY SEARCHING TIME

To evaluate the time consumption of *KVSharer*, we also test the time required for the most time-consuming part of the algorithm, Strategy Searching, as shown in Figure 4. The results show that searching for a sharing strategy on the models takes approximately one minute or less. This is expected, as Strategy Searching only requires the model to perform several inferences on a calibration dataset consisting of a few to several dozen sentences, a process that can be completed within minutes on a GPU. Note that our sharing strategy is general rather than task-specific, allowing for only one search per model, which significantly reduces the time required.

### 4.6 COMPATIBILITY WITH INTRA-LAYER COMPRESSION

Since *KVSharer* is a layer-wise KV cache compression method, it is inherently orthogonal to intra-layer KV cache techniques. Therefore, we explore the effectiveness of combining it with existing intra-layer KV cache methods. Specifically, we combine it with H2O (Zhang et al., 2024b) and PyramidInfer (Yang et al., 2024b), which are popular intra-layer compression methods. We conduct experiments on Llama2-7B and Llama2-13B, first using *KVSharer* to identify 8 layers for shared KV cache, effectively calculating the KV cache for only 24 out of the 32 layers. Then, these two layer-wise compression methods are further applied for additional 20% compression. The reproduction of PyramidInfer and H2O can be found in the Appendix B. We present the changes in PPL after adding H2O and PyramidInfer in Figure 5. At 12.5% and 25% *KVSharer* compression rates, both methods cause only a slight increase in PPL. The impact of PyramidInfer on PPL is lower compared to H2O, which is expected since PyramidInfer generally maintains better model performance.

This Figure 5 also shows the PPL of the InternLM2 and Llama2 series under different *KVSharer* compression rates, where the PPL is typically below 15, or even 10, at compression rates within 25%, allowing the model to maintain good generation quality. We present some case studies of the model's generation results in the Appendix C Table 9.

### 4.7 MEMORY COST & INFERENCE SPEED

In this section, we aim to explore the memory savings and the impact on inference speed brought by *KVSharer*. Specifically, we test the memory consumption, prefill time, and generation speed of Llama2-13B-Chat under the following settings: Full KV cache, KVSharer with 25% compression, KVSharer with 25% compression + H2O, and KVSharer with 25% compression + PyramidInfer, across different input and maximum output lengths. We show the results in Table 2.

Table 2: Memory usage (MB), prefill time (s) and generation speed (tokens/s) of the Llama2-13B-Chat. "SeqLen." represents the "input length" + "maximum output length".

| | | 512+32 | 256+2048 | 512+2048 | 1024+4096 |
|---|---|---|---|---|---|
| Full | SeqLen. | 512+32 | 256+2048 | 512+2048 | 1024+4096 |
| | Memory | 28461 | 36095 | 51639 | 58177 |
| | Prefill | 0.088 | 0.047 | 0.088 | 0.193 |
| | Generation | 11.0 | 18.0 | 18.2 | 18.7 |
| KVSharer (25%) | SeqLen. | 512+32 | 256+2048 | 512+2048 | 1024+4096 |
| | Memory | 28257 (99%) | 31403 (87%) | 37049 (72%) | 37231 (64%) |
| | Prefill | 0.087 | 0.046 | 0.087 | 0.191 |
| | Generation | 13.9 (×1.26) | 29.8 (×1.66) | 30.0 (×1.65) | 28.7 (×1.53) |
| KVSharer (25%) + H2O | SeqLen. | 512+32 | 256+2048 | 512+2048 | 1024+4096 |
| | Memory | 24852 (87%) | 26195 (73%) | 30891 (60%) | 31591 (54%) |
| | Prefill | 0.090 | 0.044 | 0.089 | 0.190 |
| | Generation | 14.1 (×1.28) | 29.2 (×1.62) | 28.3 (×1.55) | 27.1 (×1.45) |
| KVSharer (25%) + Pyr. | SeqLen. | 512+32 | 256+2048 | 512+2048 | 1024+4096 |
| | Memory | 23195 (81%) | 26059 (72%) | 30141 (58%) | 31417 (54%) |
| | Prefill | 0.089 | 0.048 | 0.089 | 0.195 |
| | Generation | 14.5 (×1.31) | 33.8 (×1.88) | 34.1 (×1.87) | 33.4 (×1.79) |

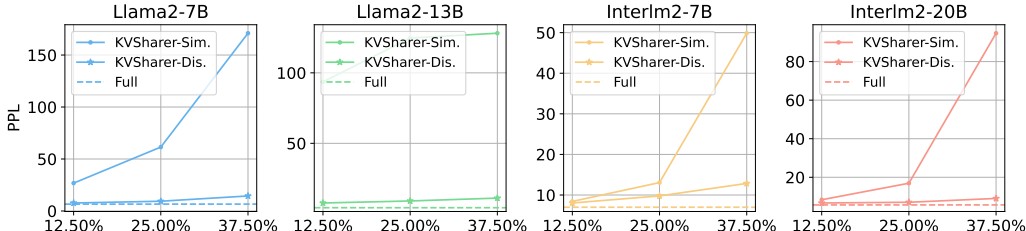

Figure 6: The model's PPL when using *KVSharer* with similarity-based sharing (+Sim.) and dissimilarity-based sharing (+Dis.). The PPL for dissimilarity-based sharing is significantly better than for similarity-based sharing.

When the sentence length is relatively short, such as 512+32 tokens, the memory-saving effect of *KVSharer* is not significant, as the current memory usage is still primarily due to the model itself. As the length increases, the memory-reducing effect begins to show. When the length reaches 256+2048 tokens, the memory savings can reach up to 30%.

In terms of speed, although there is no acceleration during the prefill phase, there is a significant acceleration during the generation phase as our results also show at least 1.2 times acceleration. When the length reaches 512+2048, it can provide over 1.6 times acceleration during the generation.

After adding PyramidInfer and H2O, the memory usage is further reduced. Additionally, Pyramid-Infer further accelerates the generation speed.

## 5 ABLATION STUDY

### 5.1 SHARING BY KV CACHE SIMILARITY OR DISSIMILARITY?

We adopt a counterintuitive sharing strategy by compressing during inference through sharing dissimilar KV cache, rather than the intuitive approach of sharing similar KV cache. This section will experimentally demonstrate that sharing based on KV cache dissimilarity performs better.

Specifically, we modify Algorithm 1 by changing the descending order based on euclidean distance to ascending order, so that KV caches are sorted from high to low similarity while keeping all other steps unchanged. We then conduct experiments on the four models used in the main experiment.

Figure 6 compares the models' PPL when sharing based on similarity versus dissimilarity. The results indicate that, for each model, at the given compression rates, the PPL of the similarity-based sharing strategy is significantly higher, often nearly twice as high or more than that of the dissimilarity-based strategy. Therefore, the method proposed in this paper is founded on sharing through dissimilarity.

Table 3: Model performance at a 25% compression rate using Wikipedia and BookCorpus as calibration dataset. For each model, using a subset of the BookCorpus dataset as the calibration dataset has little impact on *KVSharer* compared to using a subset of the Wikipedia dataset.

| LLM | Calibration Dataset | BoolQ | PIQA | HeSw | PPL |
|---|---|---|---|---|---|
| **Llama2-7B** | Wikipedia | 72.39 | 74.37 | 63.97 | 9.39 |
| | BookCorpus | 72.01 | 74.10 | 64.05 | 9.15 |
| **Llama2-13B** | Wikipedia | 78.20 | 76.71 | 72.40 | 9.11 |
| | BookCorpus | 78.34 | 76.81 | 72.18 | 9.17 |
| **InternLM2-7B** | Wikipedia | 80.37 | 79.49 | 73.22 | 9.78 |
| | BookCorpus | 80.37 | 79.49 | 73.22 | 9.78 |
| **InternLM2-20B** | Wikipedia | 80.61 | 80.96 | 75.84 | 7.05 |
| | BookCorpus | 81.08 | 80.53 | 75.46 | 7.01 |

## 5.2 EFFECT OF DIFFERENT CALIBRATION DATASETS

To investigate the impact of different calibration datasets, we replace the Wikipedia dataset with a randomly selected, equally sized subset of the BookCorpus dataset (Kiros et al., 2015). We set the compression rate to 25% and rerun the experiments, keeping all other settings unchanged.

The results are shown in Table 3. The findings indicate that using the two different calibration datasets has almost no impact on model performance, with only minimal differences in performance across several benchmarks and PPL. For InternLM2-7B, the same sharing strategy is identified with both datasets, further indicating that *KVSharer* is not sensitive to the calibration dataset. We also conduct an ablation study on calibration dataset size in Appendix A.3, Table 8, and find that the size has little impact.

Table 4: Model performance using KVSharer and random sharing strategies at a 25% compression rate.

| LLM | Strategy | BoolQ | PIQA | HeSw | PPL |
|---|---|---|---|---|---|
| **Llama2-7B** | KVSharer | 72.39 | 74.37 | 63.97 | 9.39 |
| | Random | 50.67 | 59.15 | 44.97 | 21.29 |
| **Llama2-13B** | KVSharer | 78.20 | 76.71 | 72.40 | 9.11 |
| | Random | 40.69 | 51.21 | 42.99 | 51.41 |
| **InternLM2-7B** | KVSharer | 80.37 | 79.49 | 73.22 | 9.78 |
| | Random | 63.33 | 61.73 | 58.13 | 13.58 |
| **InternLM2-20B** | KVSharer | 80.61 | 80.96 | 75.84 | 7.05 |
| | Random | 61.43 | 64.11 | 58.39 | 18.50 |

## 5.3 RANDOM SHARING V.S. KVSHARER

*KVSharer* compresses KV cache through a highly counterintuitive strategy of sharing dissimilar KV caches, which leads us to explore whether KV caches can be shared arbitrarily to achieve compression effect. Thus, we conduct comparative experiments. Specifically, we randomly select some layers' KV caches to replace others, set the compression rate to 25%, keep other settings unchanged, and evaluate the models' performance on multiple benchmarks and their PPL. We repeat the experiments three times and take the average of the results.

We present the results in Table 4. The results indicate that, compared to KVSharer's PPL of under 10, the randomly selected sharing strategy causes a significant increase in the model's PPL, reaching as high as 50 for Llama2-13B.

Across different benchmarks, the randomly selected strategy also reduces the model's performance, typically by about 30%. This set of experiments demonstrates that a randomly selected sharing strategy cannot maintain model performance, while *KVSharer*, with its search-based approach, can find a more effective sharing strategy.

However, the results also contain some surprising findings. In the case of randomly sharing the KV cache, the model's performance does not drop to zero, and the PPL does not explode to over a hundred. This suggests that there may be redundancy in the KV cache, or that the impact of the self-attention keys and values on the subsequent hidden-state calculations is not as significant as we initially thought. We will continue to explore this in the future.

Table 5: Comparison of performance on different benchmarks and PPL between Chat and Base versions of the models at the same compression rate.

| LLM | Llama2-7B | | | | Llama2-13B | | | | InternLM2-7B | | | | InternLM2-20B | | | |
|---|---|---|---|---|---|---|---|---|---|---|---|---|---|---|---|---|
| Version | Base | | Chat | | Base | | Chat | | Base | | Chat | | Base | | Chat | |
| Layer | 32 | 24 | 32 | 24 | 40 | 30 | 40 | 30 | 32 | 24 | 32 | 24 | 48 | 36 | 48 | 36 |
| BoolQ | 70.67 | 69.27 | 70.67 | 72.39 | 71.50 | 65.63 | 81.56 | 78.20 | 71.28 | 70.40 | 83.21 | 80.37 | 65.44 | 54.04 | 81.71 | 80.61 |
| PIQA | 78.18 | 76.66 | 78.18 | 74.37 | 79.71 | 75.35 | 78.24 | 76.71 | 80.30 | 79.00 | 79.60 | 79.49 | 82.10 | 81.23 | 81.39 | 80.96 |
| HeSW | 71.28 | 69.43 | 71.35 | 63.97 | 74.83 | 67.81 | 75.41 | 72.40 | 73.43 | 72.46 | 73.30 | 73.22 | 75.46 | 74.99 | 76.57 | 75.84 |
| PPL | 5.25 | 11.13 | 6.62 | 9.39 | 4.32 | 7.73 | 5.99 | 9.11 | 7.27 | 10.59 | 6.99 | 9.78 | 5.13 | 7.38 | 5.67 | 7.05 |

## 5.4 EFFECT OF KVSHARER ON DIFFERENT MODEL VERSIONS

Since the models used in our main experiments are all Chat versions, we also want to explore whether *KVSharer* can be effective on the Base versions of the models. We conduct comparative experiments using the Base versions of different models, setting the compression rate at 25%, and also comparing the results with those of the full KV cache.

We show the results in the Table 5. As shown in the result, *KVSharer* also works for Base models, as it similarly maintains a minor impact on both various tasks and PPL, comparable to its effect on the Chat model. This also demonstrates that *KVSharer* has strong generalizability.

## 6 CONCLUSION

In this paper, we introduce *KVSharer*, a layer-wise KV cache sharing method designed for efficient LLM inference. By counterintuitively sharing dissimilar KV caches, *KVSharer* reduces memory usage and boosts prefill speed during inference. Our experiments show that *KVSharer* maintains over 90% of the original performance of mainstream LLMs while reducing KV cache computation by 30%. It can also provide at least 1.3 times acceleration in generation. Additionally, *KVSharer* can be integrated with existing intra-layer KV cache compression methods to achieve even greater memory savings and faster inference. We also explore the effectiveness of the dissimilarity-based sharing approach and perform ablation studies on several components of the method.

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

# A SUPPLEMENTARY RESULTS

## A.1 MAIN RESULT

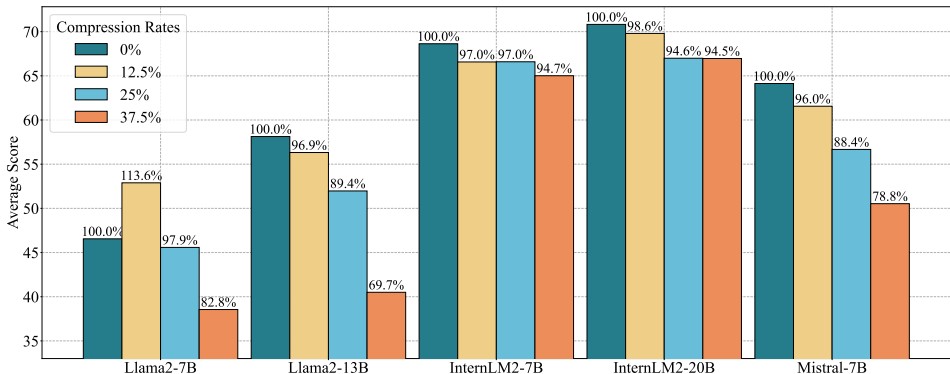

Figure 7: The percentage of the model's average score at different compression rates relative to the full KV cache model.

Table 6: The main results of our experiments. "Layer" represents the number of layers where the KV cache is actually computed.

| LLM | Layer | Reasoning | | | Language | | | Knowledge | | Examination | | Understanding | | | |
|---|---|---|---|---|---|---|---|---|---|---|---|---|---|---|---|
| | | CMNLI | HeSw | PIQA | CHID | WSC$_P$ | WSC$_G$ | CSQA | BoolQ | MMLU | CMMLU | Race$_H$ | Race$_M$ | XSum | C3 |
| **Llama2 -7B** | 32 | 32.98 | 71.35 | 78.18 | 46.04 | 37.50 | 38.46 | 66.67 | 70.67 | 45.92 | 31.86 | 35.51 | 33.15 | 19.68 | 43.78 |
| | 28 | 35.11 | 70.37 | 76.71 | 42.08 | 63.46 | 57.69 | 69.62 | 74.10 | 38.63 | 33.74 | 53.95 | 55.92 | 23.24 | 45.81 |
| | 24 | 34.89 | 63.97 | 74.37 | 37.62 | 55.77 | 59.62 | 48.65 | 72.39 | 38.38 | 27.87 | 30.33 | 31.27 | 21.30 | 41.70 |
| | 20 | 34.49 | 55.11 | 71.44 | 32.18 | 52.61 | 30.77 | 48.65 | 59.14 | 28.46 | 25.42 | 22.81 | 23.19 | 16.81 | 38.68 |
| **Llama2 -13B** | 40 | 35.06 | 75.41 | 78.24 | 48.02 | 66.35 | 67.31 | 69.78 | 81.56 | 54.64 | 38.71 | 58.46 | 64.07 | 25.84 | 50.30 |
| | 35 | 34.27 | 72.84 | 76.82 | 46.04 | 63.46 | 62.50 | 68.71 | 80.40 | 53.87 | 38.44 | 58.18 | 64.14 | 20.30 | 48.44 |
| | 30 | 34.93 | 72.40 | 76.71 | 44.06 | 53.85 | 44.23 | 69.12 | 78.20 | 53.88 | 38.19 | 53.60 | 60.45 | 0.71 | 47.29 |
| | 25 | 34.93 | 64.07 | 73.39 | 33.17 | 58.65 | 39.42 | 39.80 | 64.98 | 40.81 | 29.97 | 25.13 | 25.00 | 0.04 | 37.70 |
| **Intern. -7B** | 32 | 33.09 | 73.30 | 79.60 | 82.18 | 61.54 | 70.19 | 69.53 | 83.21 | 65.98 | 62.94 | 84.19 | 89.00 | 33.56 | 72.55 |
| | 28 | 33.07 | 72.64 | 76.71 | 83.66 | 51.92 | 65.38 | 69.70 | 80.95 | 58.12 | 62.40 | 83.68 | 89.00 | 32.43 | 72.33 |
| | 24 | 33.87 | 73.22 | 79.49 | 81.68 | 45.19 | 69.23 | 68.96 | 80.37 | 63.11 | 62.29 | 83.33 | 88.72 | 30.62 | 72.16 |
| | 20 | 33.44 | 72.23 | 77.64 | 78.71 | 42.31 | 70.19 | 68.47 | 80.09 | 63.27 | 61.81 | 80.96 | 86.84 | 25.14 | 69.10 |
| **Intern. -20B** | 48 | 54.01 | 76.57 | 81.39 | 86.63 | 50.00 | 65.38 | 74.05 | 81.71 | 66.55 | 65.98 | 86.51 | 90.25 | 33.04 | 79.51 |
| | 42 | 50.14 | 76.17 | 80.74 | 85.15 | 50.00 | 65.38 | 73.59 | 81.19 | 66.17 | 65.70 | 86.48 | 90.39 | 26.63 | 79.51 |
| | 36 | 43.65 | 75.84 | 80.96 | 84.16 | 56.73 | 55.77 | 74.20 | 80.61 | 65.98 | 64.92 | 86.13 | 90.60 | 26.47 | 79.84 |
| | 30 | 43.98 | 75.89 | 79.87 | 83.66 | 42.31 | 52.88 | 72.73 | 82.08 | 65.32 | 64.82 | 86.11 | 90.67 | 17.48 | 79.67 |
| **Mistral -7B** | 32 | 32.99 | 78.59 | 82.75 | 48.51 | 67.31 | 72.45 | 74.86 | 83.21 | 62.62 | 44.37 | 75.30 | 79.25 | 34.59 | 60.99 |
| | 28 | 32.99 | 78.87 | 81.34 | 47.03 | 57.69 | 68.91 | 73.55 | 81.16 | 58.21 | 41.83 | 71.73 | 77.09 | 31.38 | 60.00 |
| | 24 | 32.99 | 76.07 | 80.79 | 47.52 | 36.54 | 66.39 | 73.55 | 78.81 | 52.61 | 37.85 | 57.66 | 62.19 | 30.36 | 60.00 |
| | 20 | 32.99 | 73.62 | 77.58 | 47.52 | 36.54 | 65.10 | 66.99 | 76.02 | 41.06 | 29.94 | 41.02 | 44.99 | 28.63 | 45.42 |

## A.2 EXPERIMENTS ON LARGE-SIZE LLMS

Due to limitations in computational resources, we only validate the effectiveness of *KVSharer* on a subset of benchmarks and using PPL on the Llama2-70B model as shown in Table 7. We set the compression rates to 12.5% and 25%, and find that *KVSharer* effectively maintains most of the model's performance.

Table 7: The model performance achieved by applying *KVSharer* with different compression rates on Llama2-70B.

| LLM | Layer | BoolQ | PIQA | HeSw | PPL |
|---|---|---|---|---|---|
| **Llama2-70B** | 80 | 86.45 | 79.61 | 78.49 | 4.25 |
| | 70 | 84.59 | 76.93 | 77.01 | 5.59 |
| | 60 | 83.73 | 75.11 | 75.57 | 7.01 |

## A.3 ABLATION STUDY ON CALIBRATION DATASET SIZE

Table 8: Ablation study on calibration dataset size conducted on Llama2-7B under 25% compression rate.

| LLM | Calibration Dataset Size | BoolQ | PIQA | HeSw | PPL |
|---|---|---|---|---|---|
| **Llama2-7B** | 10 | 72.01 | 74.21 | 63.54 | 9.48 |
| | 30 | 72.39 | 74.37 | 63.97 | 9.39 |
| | 50 | 72.41 | 74.00 | 63.98 | 9.33 |

As shown in Table 8, the impact of calibration dataset size on *KVSharer* is also minimal, as the model still maintains good performance under a 25% compression rate. To mitigate the potential risk of obtaining suboptimal sharing strategies due to a smaller calibration dataset size, we recommend using a larger size.

## B DETAILS OF REPRODUCTION

For H2O [3] and PyramidInfer [4], we integrate their official code with our *KVSharer*. Specifically, we sparsify the KV caches for each layer sequentially according to their methods. If a particular layer's KV cache needs to utilize the sparsified KV cache from a previous layer based on *KVSharer*, we directly place the sparsified KV cache from that previous layer into the current layer. This process is used during both the strategy searching phase and the inference phase of KV sharing in *KVSharer*.

We first tune their respective hyperparameters on the full attention model to achieve approximately 20% compression rate, and then directly apply these hyperparameters to their combination with *KVSharer*.

## C CASE STUDY

---

[3] https://github.com/FMInference/H2O
[4] https://github.com/mutonix/pyramidinfer

Table 9: Under a 25% *KVSharer* compression rate, different models respond based on the prompts. The results show that the model's responses remain fluent and meaningful.

| Prompt |
| --- |
| Hello, please give a brief introduction to the Earth. |

| Llama2-7B |
| --- |
| The planet Earth is the third planet from the sun in our solar system and the only known inhabited world. With a mass of approximately 5.97 x 1worlds, Earth has a radius of about 6,311 kilometers at the equator and an average surface temperature of around 15 degrees CelsiVCentered on the axis of rotation, Earth rotates on its axis every 24 hours, which means it takes approximately 24 hours for the planet to rotate once on its axis. The planet orbits the sun at an average distance of about 149.6 million kilometers every 365 days. |

| Llama2-13B |
| --- |
| The Earth is the third planet from the Sun and the fifth largest of the eight planets in our solar system. It is a terrestrial planet, meaning it is made up of rock and metal, and has a solid surface that is suitable for life. The Earth is approximately 93 million miles (150 million kilometers) away from the Sun and takes 365.25 days to complete one orbit around the star. The planet is composed of three main layers: the crust, the mantle, and the core. The crust is the outermost layer and is made up of a variety of minerals such as granite and basalt. The mantle is the largest layer and is made up of hot, viscous rock that can flow over long periods of time. The core is divided into two layers: the inner core is solid and made up of iron and nickel, while the outer core is molten and also made up of iron and nickel. |

| InternLM2-7B |
| --- |
| The earth is an oblate spheroidal planet in our solar system and third from out of five known as "terrestrial" (rocky) planets. It's not quite round; it 'wobbles' slightly on its axis which we call precession or wobble for short! |
| The two main features that you can see with your own eyes are: – A day/night cycle this means one side will be lighted at all times when they're facing their star while other parts have no direct sunlight but do get some indirect illumination through scattering by gasses & dust. This also has implications about what timezones people use around here! |

| InternLM2-20B |
| --- |
| The earth is our home and it's where we live on this planet that has been around for about four billion years or so according to scientists who study these things called geologists (people with degrees from universities). It's made up of land masses known as continents which are separated by large bodies of water such as oceans like Pacific Ocean; there are also smaller seas including the Mediterranean Sea. |
| There have always existed life forms living here but they were not human beings until millions upon millions year ago when early humans first appeared out into open air after evolving over time through a natural selection process involving genetic mutations occurring randomly throughout history leading them towards becoming more advanced than their earlier ancestors had ever dreamed possible! |

