# OpenReview forum: "KVSharer: Efficient Inference via Layer-Wise Dissimilar KV Cache Sharing"
_ICLR.cc/2025/Conference — ICLR 2025 Conference Withdrawn Submission_

### Official Review · Reviewer_fUgR · 2024-10-24

**Soundness:** 3
**Presentation:** 3
**Contribution:** 3
**Rating:** 5
**Confidence:** 3

**Summary:**

This paper proposes a novel approach to sharing the key-value (KV) cache across different layers in a new dimension, which can lead to more efficient memory usage and improved performance.

**Strengths:**

This idea offers new insights into how memory size can be further reduced, potentially leading to more efficient model deployments and optimized hardware utilization.

**Weaknesses:**

1) The paper lacks a comparison with other cache-sharing methods, which would provide a clearer understanding of its advantages.

2) It should consider the scenario when the KV cache is quantized, as quantization is often used during inference to save energy.

3) The paper also lacks a scalability analysis, which is crucial for evaluating how well the proposed method performs as model size and complexity increase.

**Questions:**

What is the time scalability of the proposed approach? Will the inference time remain acceptable when scaling up to models with over 400 billion parameters? It would be valuable to provide an estimation or analysis to address this concern.

---

> ### Author Response · Authors · 2024-11-25
> **Response to Reviewer fUgR**
>
> Thanks for your review. We address your concerns accordingly.
>
> **C#1: The paper lacks a comparison with other cache-sharing methods.**
>
> As described in our related work section, layer-wise KV cache compression is currently rare, and existing methods require training, while ours is training-free, making direct comparisons less reasonable. We are searching for baselines and will consider introducing them in the future version.
>
> **C#2: It should consider the scenario when the KV cache is quantized.**
>
> We conducted additional experiments using the GPTQ-quantized Llama2-7B-Chat model:
>
> | Model                        | Compression Layers | PPL    |
> |------------------------------|--------|--------|
> | Llama-2-7B-Chat-GPTQ         | 0      | 8.61  |
> | Llama-2-7B-Chat-GPTQ         | 4      | 10.40  |
> | Llama-2-7B-Chat-GPTQ         | 8      | 16.68 |
> | Llama-2-7B-Chat-GPTQ         | 12     | 25.89  |
>
> We also find that KVSharer does not significantly increase the model's PPL within a 25% compression rate, further demonstrating its effectiveness. We will include this result in future versions.
>
> **C#3: The paper also lacks a scalability analysis, which is crucial for evaluating how well the proposed method performs as model size and complexity increase.**
>
> We have validated our method across multiple model families, such as the Llama2 and InternLM series, and on mainstream model sizes ranging from 7B, 13B, and 20B to 70B. The consistent conclusions can demonstrate the effectiveness of our approach.
>
> **Q#1: What is the time scalability of the proposed approach? Will the inference time remain acceptable when scaling up to models with over 400 billion parameters? It would be valuable to provide an estimation or analysis to address this concern.**
>
> We have validated our method on multiple model families and mainstream model sizes. Our approach has been proven to significantly accelerate inference speed. Conducting experiments on a 400B model exceeds our hardware capacity, requiring 1000GB or more of memory, which is beyond the reach of most researchers. Additionally, 400B models are rarely used in practice, making such demands uncommon. Our existing experiments already demonstrate the effectiveness and stability of our method.

---

### Official Review · Reviewer_fzUL · 2024-11-01

**Soundness:** 2
**Presentation:** 3
**Contribution:** 3
**Rating:** 5
**Confidence:** 4

**Summary:**

This paper introduces KVSharer, a post-training method for layerwise KV cache sharing. Based on the counterintuitive observation that sharing KV caches between layers with dissimilar, rather than similar, KV caches leads to less performance degradation, KVSharer employs a systematic search strategy for KV sharing. As a result, KVSharer reduces GPU memory consumption while maintaining model performance.

**Strengths:**

* Does not require training
* Provides an interesting and novel insight that sharing dissimilar KV caches yields better performance.
* Offers diverse and insightful evaluation results.

**Weaknesses:**

* Results show a noticeable performance drop even at low compression rates (e.g., 12.5%, 25%), which may limit the practicality of the method.
* Lacks an explanation for why sharing dissimilar KV caches yields better performance, leaving an essential aspect of the method's effectiveness rather unclear.

**Questions:**

* Why is it better to share dissimilar KV caches? Since the authors themselves describe this as counterintuitive, providing an explanation for this phenomenon would be highly valuable for the community.

* What happens if KVSharer is unable to find $C$ pairs of layers to share KV caches while satisfying the threshold $T$? It would be helpful to include a guideline on setting this threshold and any evaluation showing its impact on search performance.

* In Table 2, why does the memory usage reduction exceed the compression rate? Additionally, what is the source of the observed increase in generation throughput? Since KV cache sharing reduces memory usage but likely not memory bandwidth, it is unclear how this improves inference throughput.

---

> ### Author Response · Authors · 2024-11-25
> **Response to Reviewer fzUL**
>
> Thanks for your review. We address your concerns accordingly.
>
> **C#1: Results show a noticeable performance drop even at low compression rates (e.g., 12.5%, 25%), which may limit the practicality of the method.**
>
> Maintaining over 90% of the model's performance with a 25% compression rate is sufficient for many use cases. Additionally, as shown in our case study on page 16, the responses generated by our model are both fluent and knowledgeable, which can meet the requirements of various scenarios.
>
> **C#2&Q#1: Lacks an explanation for why sharing dissimilar KV caches yields better performance, leaving an essential aspect of the method's effectiveness rather unclear.**
>
> We are working on providing both theoretical and empirical evidence. However, the KVSharer proposed in this manuscript effectively reduces memory usage while maintaining high performance, and its counterintuitive findings offer insights for future model improvements, highlighting our contribution.
>
> **C#3: What happens if KVSharer is unable to find C pairs of layers to share KV caches while satisfying the threshold T?**
>
> As noted in the footnote of page 6, this phenomenon does not occur when the threshold is set to a reasonable value, such as our recommended 0.5.
>
> **Q#2: Why does the memory usage reduction exceed the compression rate? What is the source of the observed increase in generation throughput?**
>
> For memory reduction, we speculate that it might be due to PyTorch's underlying mechanisms utilizing fragmented memory more efficiently. Of course, we will conduct a more in-depth analysis. As for inference speed, we suspect that the acceleration could be attributed to reduced memory read/write operations, even though the computation load hasn't decreased. We are also exploring this further.

---

### Official Review · Reviewer_jb9o · 2024-11-02

**Soundness:** 1
**Presentation:** 2
**Contribution:** 1
**Rating:** 5
**Confidence:** 4

**Summary:**

This paper introduces KVSharer, a plug-and-play method for compressing the key-value (KV) cache of large language models (LLMs) during inference. Unlike the intuitive approach of sharing similar KV caches, KVSharer is based on a counterintuitive observation: sharing different KV caches across layers does not significantly degrade model performance. KVSharer employs a search strategy to identify the optimal KV cache sharing policy across different layers, substantially reducing GPU memory usage while retaining most of the model’s performance. Additionally, KVSharer is compatible with existing intra-layer KV cache compression methods, offering a complementary approach to memory optimization for LLMs.

**Strengths:**

1. This paper addresses a good research topic: efficient LLM inference.

2. The paper is well-organized.

3. The proposed method is clearly presented.

**Weaknesses:**

1. **Lack of novelty and research depth:** This main technique is to share the dissimilar KV cache for efficient inference, which is quite simple. Although authors claim that this originates from a counterintuitive observation, there is no motivation provided in the methodology section. Therefore, both of the novelty and the research depth of this paper are not qualified for the top AI conference.

2. **Unreasonable observation without further analysis:** The observation that the sharing the dissimilar KV cache brings in better accuracy than sharing the similar one sounds unreasonable, the dissimilar KV states output different attention scores, making the LLM attend to different part of the query token. It is more convincing that the obtained conclusion is just a coincidence and varies across the models and datasets, considering that no in-depth analysis has been provided.

3. Lack Needle-in-a-Haystack experiment.

**Questions:**

See Above.

---

> ### Author Response · Authors · 2024-11-25
> **Response to Reviewer jb9o**
>
> Thanks for your review. We address your concerns accordingly.
>
> **C#1: Lack of novelty and research depth: no motivation provided in the methodology section.**
>
> We respectfully disagree with your viewpoint. Our method is derived from experimental observations, as described in Lines 064-067. While we currently cannot provide a theoretical proof, we have validated our observations through extensive experiments across various models. Moreover, many existing works focus on sharing based on parameter or representation similarity. In contrast, our counterintuitive findings offer a more novel perspective.
>
> **C#2: Unreasonable observation without further analysis: The observation sounds unreasonable and the obtained conclusion is just a coincidence and varies across the models and datasets.**
>
> We also respectfully disagree with your viewpoint. We have obtained consistent conclusions across a wide range of datasets and models, and our results are certainly not coincidental. While we will attempt to conduct deeper analyses in future versions, we do not accept the doubts about our experimental results.
>
> **C#3: Lack Needle-in-a-Haystack experiment.**
>
> We will include the experiments on related long-context benchmarks in the future version.

---

> > ### Comment · Reviewer_5E3J · 2024-11-28
> >
> > Thank you for your reply! I hope the review will help make this paper better in the future!

---

> ### Comment · Reviewer_jb9o · 2024-12-01
>
> Thanks for the comments. I have raised my score and hope the author could revise the paper based on all the reviewers' suggestions in the future. Thank you.

---

### Official Review · Reviewer_5E3J · 2024-11-03

**Soundness:** 2
**Presentation:** 3
**Contribution:** 2
**Rating:** 3
**Confidence:** 4

**Summary:**

This paper first presents a counterintuitive phenomenon when attempting to leverage the cross-layer pattern to improve the efficiency of the LLM generative inference computation, where sharing dissimilar KV caches better preserves the model performance. Based on this observation, this paper introduces a method named KVSharer, which integrates this observation to implement efficient cross-layer KV cache sharing. An empirical study has been conducted to verify the effectiveness of the proposed methods.

**Strengths:**

- S1. This paper explores an important problem of improving the efficiency in utilizing KV cache in LLM generative inference.

- S2. The related work and research context are well summarized.

**Weaknesses:**

- W1. Heuristic-based on aggregated information. As enumerated in Section 3.1.2, the proposed method uses the averaged value of the KV-cache to consider the similarity between different layers -- it is a little confusing why such highly integrated information could guide the sharing policy, considering lots of recent work has been exploring the KV-cache utilization at token, layer, and head level jointly. My concern is whether such a highly aggregated metric is informative or not.

- W2. My main concern is about the experimental setup. There is a significant mismatch between the motivation example in the introduction, e.g., "During the LLM inference phase, the KV cache typically accounts for 80% of the total memory usage."  and the benchmarked settings, where the context window is set to just a few thousand, e.g., up to 1024+4096 in Table-2. Unless batching to an extremely large value (not mentioned in the paper), there is a significant gap between the motivation and the experiments. I think it would be critical to evaluate the performance of the proposed method over long-context benchmarks (e.g., Infiniti-bench) where the model's context window should be from 32K to 128K  (or even longer). Otherwise, the truly useful scenario is not evaluated.

**Questions:**

Please address the corresponding concern listed in the weakness section.

**Details Of Ethics Concerns:**

Not applicable.

---

> ### Author Response · Authors · 2024-11-25
> **Response to Reviewer 5E3J**
>
> Thanks for your feedback. We will address your concerns as followed:
>
> **Q#1: Whether such a highly aggregated metric(the averaged value of the KV-cache) is informative or not.**
>
> Since previous work rarely considered layer-level KV cache compression, as you mentioned, they mainly focused on token- or head-level KV cache representations. However, our work requires comparing the similarity between entire layers. To preserve the layer-level KV cache representation for each sample in the calibration dataset as much as possible, it is reasonable to average the layer-level KV cache representations across all samples without bias toward any particular sample. Moreover, this averaging approach is very common and is frequently used in prior work to generate hidden states or heatmaps for attention maps.
>
> **C#1: It would be critical to evaluate the performance of the proposed method over long-context benchmarks.**
>
> Thank you for your suggestion. We will include experiments on long-context benchmarks in the future version.

---

### Official Review · Reviewer_fCuJ · 2024-11-04

**Soundness:** 1
**Presentation:** 3
**Contribution:** 2
**Rating:** 3
**Confidence:** 4

**Summary:**

This paper introduces a new inter-layer KV cache compression technique through layer-wise KV cache dis-similarity search and sharing. The layers are ranked pairwise in accordance with their dis-similarity score. For each pair, an earlier layer's KV will be shared and reused by an later layer for efficient pre-filling and generation.

**Strengths:**

This paper and the technique introduced have the following strengths:

1. Paper writing is easy to follow with good figures and illustrations.
2. The experiment sections demonstrate KVSharer can be used in orthogonal with other intra-layer KV compression techniques like H2O and PyramidInfer to achieve higher memory saving and more significant speedup.
3. The paper brings up a new angle

**Weaknesses:**

I have several concerns about the paper:

1. Even though layer pairs are ranked from high dis-similarity to low dis-similarity, whether to use the pair still depends on the cosine similarity between the the KV-cache compression model and the original model. There is a possibility that the cosine similarity check, rather than dis-similarity ranking, plays a major role.

2. A major claim in the paper is dis-similarity metrics is better than similarity metrics when it comes to inter-layer KV cache sharing. Empirical evidences are provided in Section 5.1 and Figure 6 when changing the Euclidean-distance based ranking from descending order (dis-similarity) to ascending order (similarity). However, I didn't find any theoretical and empirical evidence that "Euclidean distance for KV cache is a sufficient good metrics" in comparison with the other SOTAs. More specifically, how does KVSharer compare with other layer-wise compression strategies, for example miniCache [1], LCA [2], CLLA [3] and simpleLayerKV [4]? Without the experiment results, I don't think the paper is ready at this stage for publication.

[1] Liu, Akide, et al. "MiniCache: KV Cache Compression in Depth Dimension for Large Language Models." arXiv preprint arXiv:2405.14366 (2024).

[2] Brandon, William, et al. "Reducing Transformer Key-Value Cache Size with Cross-Layer Attention." arXiv preprint arXiv:2405.12981 (2024).

[3] Yang, Zhen, et al. "Lossless KV Cache Compression to 2%." arXiv preprint arXiv:2410.15252 (2024).

[4] Zhang, Xuan, et al. "SimLayerKV: A Simple Framework for Layer-Level KV Cache Reduction." arXiv preprint arXiv:2410.13846 (2024).

**Questions:**

I think the paper will be much more ready if the authors could address the following questions (from high to low priority):

1. Could the authors provide comparisons with other layer-wise compression strategies in terms of accuracy and system performances?

2. Did the authors investigate the relationship between dis-similarity ranking and the acceptance rate by the thresholding condition? It's possible that the cosine similarity check, rather than dissimilarity ranking, plays a primary role. In principle, if the "higher dis-similarity --> inter-layer KV cache sharing gives better performance" hypothesis holds, then a higher rank should correspond to a higher acceptance rate. Could the authors provide additional results and justification on this point?

3. There is an important threshold in this work: cos-similarity (representation similarity) threshold that determines whether to accept a KV cache pair, Can the authors provide explanations on how the value is determined/searched? Moreover, the number of target shared KV cache layers is also an important hyper-parameter, and this is discussed in the paper an ablation study on in Table 1. But can the authors provide some guidance/calculation on how this number translate to memory saving and inference speedup?

4. For KV cache dissimilarity distance, why did the authors choose Euclidean distance? Could the authors ablate on other distance metrics? Similarly, for cosine similarity from the final layer hidden states, what if some other metrics like angular distance is used (less important, just wondering)?

---

> ### Author Response · Authors · 2024-11-25
> **Response to Reviewer fCuJ**
>
> Thanks for your comments. We address your concerns as follows.
>
> **C#1&Q1: There is a possibility that the cosine similarity check, rather than dis-similarity ranking, plays a major role. Did the authors investigate the relationship between dis-similarity ranking and the acceptance rate by the thresholding condition.**
>
> When preparing the manuscript, we have found that KVSharer selects pairs mostly ranked within the top 30% for dissimilarity, confirming it uses dissimilarity for KV cache sharing rather than cosine similarity check. We will include this analysis in the future version.
>
> **C#2&Q#2: No theoretical and empirical evidence that "Euclidean distance for KV cache is a sufficient good metrics". Why did the authors choose Euclidean distance? Could the authors ablate on other distance metrics?**
>
> We chose to use Euclidean distance based on our experimental findings, and we also compared it with cosine similarity while preparing this manuscript. We conducted experiments on Llama2-7B-Chat, the following are the supplementary results:
>
> | Metric          |   Similar    | Layers | PPL   |
> |------------------|-----|---|-------|
> | Cosine | Similarity  | 4      | 8.96  |
> | Cosine | Dissimilarity | 4      | 8.57  |
> | Cosine | Similarity  | 8      | 15.68 |
> | Cosine | Dissimilarity | 8      | 15.11 |
> | Cosine | Similarity  | 12     | 42.81 |
> | Cosine | Dissimilarity | 12     | 30.67 |
>
> The experimental results indicate that when cosine similarity is used instead of Euclidean distance similarity, the observed pattern also remains consistent: leveraging dissimilarity for sharing performs better than using similarity for sharing. Moreover, since models with the same compression rate achieve better perplexity (PPL) when using Euclidean distance for sharing compared to cosine similarity (as shown in Figure 5 of the manuscript), we chose to use Euclidean similarity as the metric. We will include these analyses in the future version.
>
> **C#3&Q#3: Provide comparisons with other layer-wise compression strategies.**
>
> The methods in [2], [3], and [4] you mentioned all require post-training to be utilized, whereas our KVSharer is training-free, making a direct comparison less necessary. Additionally, [3] and [4] were public after the ICLR submission deadline, so they are not relevant for comparison in this manuscript. Furthermore, [4] is not strictly a layer-wise compression strategy; it focuses on dropping tokens within the KV cache of certain layers.
> We have already discussed [1] and [2] in the related work section. However, [1] lacks publicly available code, and we are actively working on reproducing it. The results will be included in the future version.
>
> **Q#4: Can the authors provide explanations on how the cos-similarity (representation similarity) threshold is determined/searched?**
>
> As noted in the footnote of page 6: During strategy searching, the similarity of the last layer's hidden state between the compressed and original models is usually above 0.8. A 0.5 threshold is set to avoid rare cases of output collapse. Since this is infrequent, we did not conduct an ablation study on T. And we recommend setting the threshold around 0.5.
>
> **Q#5: Provide guidance/calculation on how the number of target shared KV cache layers to memory saving and inference speedup.**
>
> As mentioned in Lines 319-320, we recommend setting the KV cache compression rate to around 25% to maintain good model performance.

---

> ### Comment · Reviewer_fCuJ · 2024-11-30
>
> For C#1&Q#1 + C#2&Q#2, I believe adding more detailed results and incorporating them into future revisions is important toward making this paper publication-ready.
>
> For C#3 & Q#3: while I understand that some relevant works have been released only recently, it is essential to benchmark the proposed approach against baseline methods that share similar goals of generation acceleration and memory savings through KV cache compression. In addition to benchmarking KVSharer against itself with different compression ratios (and w/ vs. w/o applying other intra-layer compression techniques).
>
> Regarding Q#5, my question was: "How does this number **translate** into memory savings and inference speedup?" THe reason I asked was because choosing different compression ratios is a trade off between speedup/memory saving vs. performance. I believe the paper would benefit from a more thorough system profiling. Such profiling should illustrate how system performances are affected by varying compression rates.

---

> > ### Author Response · Authors · 2024-12-01
> > **Response to Reviewer fCuJ**
> >
> > Thank you for your response and the further clarification on Q5. We will include the experiment you mentioned in the next revision to improve the manuscript. Thank you!

---

### Note · Authors · 2024-12-16

**Comment:**

We appreciate the suggestions from the AC and all reviewers, and we will refine the paper further in subsequent versions.

**Withdrawal Confirmation:**

I have read and agree with the venue's withdrawal policy on behalf of myself and my co-authors.